# Use of Hair as Matrix for Trace Elements Biomonitoring in Cattle and Roe Deer Sharing Pastures in Northern Italy

**DOI:** 10.3390/ani14152209

**Published:** 2024-07-30

**Authors:** Susanna Draghi, Nour Elhouda Fehri, Fatma Ateş, Nural Pastacı Özsobacı, Duygu Tarhan, Bengü Bilgiç, Banu Dokuzeylül, Çağla Parkan Yaramış, Alev Meltem Ercan, Mehmet Erman Or, Petra Cagnardi, Gabriele Brecchia, Giulio Curone, Federica Di Cesare

**Affiliations:** 1Department of Veterinary Medicine and Animal Sciences, University of Milan, 26900 Lodi, Italy; nour.fehri@unimi.it (N.E.F.); petra.cagnardi@unimi.it (P.C.); gabriele.brecchia@unimi.it (G.B.); federica.dicesare@unimi.it (F.D.C.); 2Department of Biophysics, Faculty of Medicine, Istanbul Beykent University, Istanbul 34398, Turkey; fatmaalkan@beykent.edu.tr; 3Department of Biophysics, Cerrahpasa Faculty of Medicine, Istanbul University-Cerrahpasa, Fatih, Istanbul 34098, Turkey; n.pastaciozsobaci@iuc.edu.tr (N.P.Ö.); meltem@iuc.edu.tr (A.M.E.); 4Department of Biophysics, School of Medicine, Bahcesehir University, Istanbul 34734, Turkey; duygu.tarhan@bau.edu.tr; 5Department of Internal Medicine, Faculty of Veterinary Medicine, Istanbul University-Cerrahpasa, Istanbul 34320, Turkey; bengu.bilgic@iuc.edu.tr (B.B.); b9eylul@iuc.edu.tr (B.D.); ermanor@iuc.edu.tr (M.E.O.); 6Department of Plant and Animal Production, Vocational School of Veterinary Medicine, Istanbul University-Cerrahpasa, Istanbul 34320, Turkey; cparkan@iuc.edu.tr

**Keywords:** bioindicators, eco-toxicology, environmental toxicology, PTEs, wildlife

## Abstract

**Simple Summary:**

This study investigates using hair analysis to monitor potentially toxic elements (PTEs) in cattle and roe deer sharing pastures in Northern Italy. PTEs include essential and non-essential elements that, if unbalanced in organisms, can lead to health issues. Hair analysis is a non-invasive method that allows retrospective evaluation of PTE exposure. Aluminum, As, Cd, Cr, Ni, Pb, Cu, Mg, Fe, and Zn were measured. Findings indicate significantly higher As levels in roe deer due to selective feeding, while Cd and Pb levels align with other studies. Cattle have lower Cu, Fe, and Zn levels, likely due to dietary differences. Elevated Cr and Ni in cattle suggest contamination or physiological differences. Hair analysis proves valuable for monitoring environmental PTE exposure, emphasizing interspecies differences and the potential of both animals as bioindicators.

**Abstract:**

Intensive cattle breeding’s environmental challenges are prompting shifts to extensive, pasture-based systems, influencing nutrient and pollutant uptake. PTEs are essential and non-essential elements, regularly found in the environment and organisms, and in which unbalances lead to health issues. Hair analysis, a non-invasive method, provides retrospective PTE exposure evaluation. This study aims to understand exposure and species-specific accumulation patterns of PTEs in cattle and roe deer sharing pastures in Northern Italy using the hair analysis. Aluminum, As, Cd, Cr, Ni, Pb, Cu, Mg, Fe, and Zn were quantified through the use of ICP-OES. Findings show As levels significantly higher in roe deer due to their selective feeding, while Cd and Pb levels align with other studies. Essential elements like Cu, Fe, and Zn are lower in cattle, possibly due to diet differences. Higher Cr and Ni levels in cattle suggest contamination or physiological differences in accumulation patterns. In conclusion, hair analysis is valuable for monitoring environmental PTE exposure, highlighting significant interspecies differences and the potential of both animals as bioindicators in shared grazing areas.

## 1. Introduction

The traditional semi-extensive farming system of cattle in Northern Italy provides the use of pastures from spring to autumn, during which the animals feed on fresh forage, and indoor housing during winter with a diet based on hay obtained from lands adjacent to pastures and farms [1]. Inevitably, concerning the nutrients and the pollutants acquired through the diet, this system creates a strong dependence on the territory in which the animals are raised [2]. Among the substances to consider in this context, particular importance is given to the essential and non-essential elements, also called potentially toxic elements (PTEs). The term PTEs is attributed to the fact that many of them are essential for carrying out physiological activities but become toxic at high concentrations; others are non-essential or toxic, but even in this case, their toxicity depends on the concentration. Due to their importance, over the decades PTEs have been extensively studied in both humans and animals, as their imbalances result in serious health issues [3]. 

The dual nature of PTEs makes it necessary to monitor their presence and quantities. Given that the absorption of any substance by a living organism depends on various intrinsic factors, both related to the animal (e.g., species, gender, or age) and the substance (chemical–physical properties), monitoring their presence and quantity in soil, water, feed, or food is insufficient to define real exposure [4]. That is why, for many years, environmental toxicology studies have employed biomonitoring. The term biomonitoring is defined as “the measurement and assessment of toxicants or their metabolites either in tissues, secreta, excreta, expired air, or any combination of these to evaluate exposure and health risk”. Since decades, wild animals have been used for this type of study [5] as considered effective tools due to their direct exposure to the natural environment [6,7]. Many wild mammals have been identified as ideal bioindicators, among which the roe deer (Capreolus capreolus) stands out [8,9]. Being an herbivorous ruminant with limited home ranges, it can effectively indicate the presence of environmental contaminants in a specific territory [10]. Considering the reasons behind its selection as an ideal bioindicator, the potential for using pasture-raised cattle as biomonitoring tools also becomes obvious. Pasture-raised cattle, with home ranges comparable to those of roe deer (16–20 hectares) [11], share a similar characteristic, i.e., they are ruminants whose diet relies on local forage. Despite differences in body size, life expectancy, and human intervention in diet, particularly feed supplementation in the case of cattle, both species could be able to offer valuable insights into PTE accumulation due to intrinsic species-specific factors when coexisting in grazing areas.

An effective method for assessing PTE levels in animals involves the examination of hair [12]. Hair, once separated from the epidermis, serves as a metabolically inert biological matrix that is chemically uniform [13]. Its non-invasive collection from both wild and domestic animals is straightforward, and its growth pattern permits a retrospective analysis of the examined element for several months post-collection [14]. Moreover, the sampling can be repeated on the same skin area, allowing for repeated measurements [2]. The hair analysis facilitates the determination of prolonged exposure to both trace elements and heavy metals, owing to the chelation capability of the sulfhydryl group (–SH) in cysteine [14].

Therefore, the purpose of this study is to employ the hair of both cattle and roe deer from the same area for biomonitoring the quantity of some PTEs, i.e., Al, As, Cd, Cr, Cu, Fe, Mg, Ni, Pb, and Zn. Secondly, it aims to compare the differences in PTE accumulation between the two species to understand the influence of the species and the rearing environment. Finally, it would determine whether roe deer exhibit accumulation patterns similar to cattle and can be used as a biomonitoring tool to assess the health of an area potentially designated for grazing.

## 2. Materials and Methods

### 2.1. Animal and Hair Collection

Before starting this study, the approval of the Institutional Animal Care and Use Committee of Università degli Studi di Milano (Permission OPBA_26_2022) as a non-experimental project was requested and obtained. 

At first, a specific area of Northern Italy with the presence of semi-extensive cattle breeding farms and a well-structured hunting activity was chosen (Figure 1). From a farm with pastures distributed throughout the sampling area, 40 healthy cows were randomly selected. All the cows were females, multiparous (average 2.2 deliveries), aged averagely 4.7 years; all animals were of dual-purpose, producing both milk and meat, raised on pasture from April to November and kept in barns from December to March. During the months they were indoors, the cows were fed with hay obtained from areas near the pastures. Throughout the year, each cow was fed a complete feed in the amount of 2.5 kg per day (the specific characteristics of the feed, i.e., ingredients, vitamins, trace elements, and analytical data, are given in Appendix A). To standardize, the 40 sampled roe deer were all females with an average age of 3 years, randomly selected from the hunting plan. The age of the animals was estimated through the evaluation of dental eruptions and erosion.

In both species, the hair samples were collected from an area of 10 cm^2^ on the left side of the animal at the level of the costal arch. For hair collection, an electric shaver was used, and the hairs were cut at the base. The sampling procedure was conducted separately for the two species. The cows’ hair was sampled directly in the pasture over multiple days to cover the entire hunting period during which roe deer hair was collected (June–October). The roe deer hair was sampled during regular hunting activities at the game meat processing center. After sampling, the hair was placed in specific plastic bags and stored at room temperature, in a dry environment, and protected from sources of light.

### 2.2. Analysis of PTEs

Aluminum (Al), arsenic (As), cadmium (Cd), chromium (Cr), nickel (Ni), lead (Pb), copper (Cu), magnesium (Mg), iron (Fe), and zinc (Zn) elements were carried out by using an inductively coupled plasma-optical emission spectrophotometer (ICP-OES; Thermo iCAP 6000series) at Istanbul University-Cerrahpasa. ICP-OES device parameters for determining PTEs are presented in Table 1. The quality assurance of ICP-OES analysis was ensured through the utilization of appropriate test solutions, each containing 2000 ppm (mg/L) for every element under examination, sourced from Chem-Lab NV. Standard solutions for all PTEs were meticulously prepared by diluting standards with 1000 ppm (mg/L) concentrations for each element, also obtained from Chem-Lab NV, in deionized water. A 3-point calibration was conducted using standard and blank solutions as reference materials. This process resulted in the acquisition of reproducible and linear calibration curves for analysis, with a determination of the correlation coefficient for each measured element. The calibration graph was generated using blank and standard solutions on the ICP-OES device, facilitating the subsequent elemental analysis of the prepared samples. To ensure precision, each measurement was replicated three times, and the results were averaged for comprehensive analysis. This rigorous methodology is aimed at maintaining the integrity and accuracy of the ICP-OES analytical process. In the study, the appropriate wavelengths (Table 2) of all elements were used for the analysis by the ICP-OES device. According to methodology, before starting the analyses, the hair was washed with acetone, three times with deionized water, and finally with acetone to remove surface impurities and any adhesive contamination present on the surface of the hair. Then, all samples underwent dissolution in a drying oven (Heraeus W.C., Hanau, Germany) at 180 °C. The average sample weight used was 0.143 g. This process involved the addition of 2 mL of 65% nitric acid (Merck, Darmstadt, Germany) and 1 mL of 60% perchloric acid (Panreac, Barcelona, Spain). After cooling to room temperature, the suspension was vortexed, and distilled water was added to the samples, reaching a total volume of 10 mL. The quantification of the concentration of each element was conducted individually, with consideration given to the weight of the respective sample. This systematic approach ensured a precise and methodical assessment of the elemental concentrations in the samples. The recovery of the analyzed quality control was between 96% and 108%. Table 3 shows the results of the ICP-OES method validation. Trace and toxic element levels were expressed as mg·kg^−1^ of sample wet weight.

### 2.3. Statistical Analysis

Statistical analyses were conducted using GraphPad InStat 8 software (version 8.0.2). Initially, the data were categorized based on species (bovine = 40 and roe deer = 40). Subsequently, the Shapiro–Wilk normality test was performed. As the data were found to be non-normally distributed, a non-parametric statistical test (Mann–Whitney test) was applied for comparisons between the two categories.

## 3. Results

During this study, statistically significant differences were highlighted between the two species in the concentration of PTEs in the hair. In cattle, compared to roe deer hair, the content of Cr, Ni, Mg, and Zn was found to be significantly higher with a *p*-value < 0.001. In the case of Al, As, Pb, and Cu, significantly higher concentrations were recorded in roe deer hair with *p* < 0.001, except for Pb, which showed *p* = 0.051. The average, median with percentiles, minimum, and maximum data are reported in Table 4. Graphical representations of comparisons between the two species and statistical significances are shown in Figure 2.

## 4. Discussion

Worldwide, hair analysis is considered a suitable method for the assessment of the health status and the mineral metabolism of animals. In this study, aluminum (Al), arsenic (As), cadmium (Cd), chromium (Cr), nickel (Ni), lead (Pb), copper (Cu), magnesium (Mg), iron (Fe), and zinc (Zn) were quantified in the hair of both cattle and roe deer grazing within the same geographical area. The aim was to use them as bioindicators for the presence and environmental quantities of PTEs and subsequently compare the concentrations between the two species to identify any interspecies differences. This approach is useful in understanding the potential physiological, ethological, and dietary influences on PTE accumulation in hair [2]. The cattle were grazed in pastures from March to November, thus sharing the home range with the roe deer hunted in that area. The use of hair to assess mineral metabolism and the health status of animals has been employed in various domestic species, such as dogs [15], cats [16], horses [17], and cattle [18], as well as in various wild species like bison [19], red deer [20], mule deer [21], and even roe deer [12]. Technically, interpreting data regarding PTE content in hair is only possible after comparison with species-specific reference values [22]. In our case, reference values exist only for cattle [23,24].

Comparing the median values identified in the hair of cattle enrolled in this study with the published reference values [23,24], both similarities and differences were revealed. The concentrations of As, Cd, and Pb turned out to be very similar compared with the reference values reported by Miroshnikov et al. [23] in Hereford cows. In the hair of roe deer, a similar study by Cygan-Szczegielniak et al. [12] conducted in Poland showed a higher concentration of Cd and a lower concentration of Pb compared to the concentrations identified in the roe deer hair in our study. In the study of Cygan-Szczegielniak et al., As has not been evaluated in roe deer hair. Comparing the concentration of As, Cd, and Pb between the two species, only As is significantly higher in roe deer hair (*p* < 0.001; Figure 2). As is a metalloid, both naturally present in the environment and derived from human activities. It is regularly traced in surface and groundwater, and its quantity is due to the mineral–water interaction [25]. Considering the natural presence of this element in rivers and streams [26], the identified difference between the two species could be due to their behavior and rearing conditions. Indeed, deer have larger home ranges and a larger number of watering points. In contrast, cattle, confined to fenced pastures with restricted access to watering points, may experience lower exposure to arsenic if the water sources have lower concentrations, hence resulting in reduced arsenic accumulation in their hair. Moreover, due to its selective browsing habits [27], the roe deer tends to target plant buds, roots, and seeds, which are arsenic-accumulating organs, thus elevating its susceptibility to arsenic uptake [28].

Regarding Cu, Fe, and Zn in cattle, the concentrations obtained in our study were approximately half compared to those obtained in the study by Miroshnikov et al. [24] on Holstein cows. These concentration differences between our and the reference values, particularly in elements considered essential for cattle, could be attributed to variations in the biogeochemical composition of pastures and dietary supplements [29]. Indeed, the cattle used in our study were dual-purpose (milk and meat) that grazed in pastures for 8 months, whereas the cattle used as reference values are Holstein cows bred for milk production and always kept in barns. Regarding the content of Cu, Zn, and Fe in roe deer hair, it was found that the concentration of Cu and Zn identified in our study was much lower compared to that identified by Cygan-Szczegielniak et al. [23], while the Fe content resulted 10 times higher compared to the study of Długaszek et al. [30]. Comparing the content of Cu and Zn between the considered species, they were at significantly higher concentrations in roe deer species. This result is rather controversial, as cattle were administered supplementary feed containing Cu and Zn, and thus higher concentrations were expected at the fur level of this species. However, one possible explanation could be attributed to the different diets. Temporarily setting aside the supplements, cattle primarily feed on fresh or dried forage, while roe deer, as reported in the literature, feed on roots, buds, and leaves of plants and shrubs [31]. In the case of Cu, a study by Yruela et al. [32] reports that the genes responsible for copper uptake, distribution, and storage in plants are expressed at the level of roots, buds, and flowers. The same principle applies to Zn, which tends to be higher in reproductive vegetative tissues as well as in shoots and roots [33].

The concentrations of Cr and Ni found in our cattle were many times higher compared to those identified in the hair of both Hereford cattle raised in pastures [23] and Holstein cows raised in barns [24]. The proposed reference interval of Cr for Hereford cattle raised in pastures is 0.053–1.121 mg·kg^−1^. In the case of Holstein Friesian cattle raised in barns, the interval narrows to 0.0671–0.409 mg·kg^−1^. In our study, the median value of Cr in bovine was 3.620 mg·kg^−1^. Chromium is considered an essential element for ruminants as it plays a role in carbohydrate and lipid metabolism and has beneficial effects on the immune system [34]. However, at excessive concentrations, it has a negative impact on the respiratory and gastrointestinal systems and also exhibits carcinogenic and embryotoxic effects [35]. Chromium is an element that is both naturally present in the Earth’s crust and originates from industrial processes [36], and its bioavailability depends on soil composition [37]. Therefore, cattle raised in different areas, grazing on different soils, or fed with forage grown in soils with varying compositions could necessarily have different tissue concentrations of Cr [18]. The concentration of Cr in roe deer hair was similar to studies conducted on the same species and other wild species [38]. 

The median Ni concentration identified in the hair in our study was 2.391 mg·kg^−1^, whereas the reference intervals were 0.108–0.595 mg·kg^−1^ for lactating cows and 0.280–1.575 mg·kg^−1^ in beef cattle [23,24]. Nickel is not considered an essential element, and its excess is carcinogenic and teratogenic [39]. Similarly for Cr, in the case of Ni, the concentration in roe deer hair was found to be consistent with the concentrations identified in other studies on the same species and on other wild species [38]. It is naturally present in the Earth’s crust and, due to its chemical and physical properties, finds widespread use in various applications, including modern metallurgies such as alloy production, electroplating, and the manufacturing of nickel-cadmium batteries [40]. The extensive use of products containing Ni inevitably contributes to environmental pollution [41] and determines the difference in exposure. 

Comparing the concentration of Cr and Ni in the hair of cattle and roe deer, it was found that cattle showed a significantly higher concentration of both elements. The sources of exposure are mainly dietary; however, the reason for this difference is unclear. It could be hypothesized contamination of cattle drinking water or supplementary feed, excluding hay, as it is collected in the same area where roe deer live. Alternatively, a physiological difference between the species such as body mass, rumen size, and feeding behavior (indeed, based on feeding behavior, cattle are grazers, while roe deer are selective browsers) could have led cattle to a higher intake of Ni and Cr. Future studies identifying the concentration of Cr and Ni in the feed of both species might help explain this difference.

Aluminum and Mg were not included in the reference values [23,24] but were identified in another study of Linhares et al. [18], where, in the hair of pasture-raised cattle, higher concentrations were found compared to our study. Our lower concentrations of Al and Mg could be probably due to the different soil composition; indeed, cattle enrolled in the study of Linhares et al. belonged to areas with volcanic soils [18]. 

The Al content identified in the hair of roe deer in this study was found to be much lower compared to that identified in the hair of other wild animals [38]; the difference could be due to intraspecific and interspecific differences, sampling period, and sampling area. For the concentration of Mg, it resulted similar to another study of our research group in roe deer hair [10]. Moreover, there are also studies reporting the concentration of Mg in the fallow deer antlers [42] or in the cranial bone and antlers of red deer [43], but considering the physiological inter-specific differences and the differences between the matrices, the comparison is not possible.

Meanwhile, comparing the concentration of Al in hair between cattle and roe deer, the hair of deer was found to have a significantly higher level. For this element as well, the main source of exposure appears to be the diet. Aluminum primarily accumulates in plant roots [44], so the feeding behavior of deer could explain the higher concentration in their fur compared to cattle. On the other hand, Mg was found to be higher in the hair of cattle, despite the supplementary feed given to them daily not containing it. Therefore, it is possible to hypothesize that this result depends on the physiology of the two species or their body mass.

## 5. Conclusions

In conclusion, this study highlights the significance of hair analysis as a valuable method for assessing the health status and mineral metabolism of animals, particularly in understanding the accumulation of PTEs. By comparing the concentrations of various elements in the hair of cattle and roe deer, significant interspecies differences were identified, shedding light on potential environmental influences, dietary habits, and physiological factors impacting element accumulation. Further research into the dietary sources of these elements and physiological differences between species is warranted to fully elucidate these findings.

## Figures and Tables

**Figure 1 animals-14-02209-f001:**
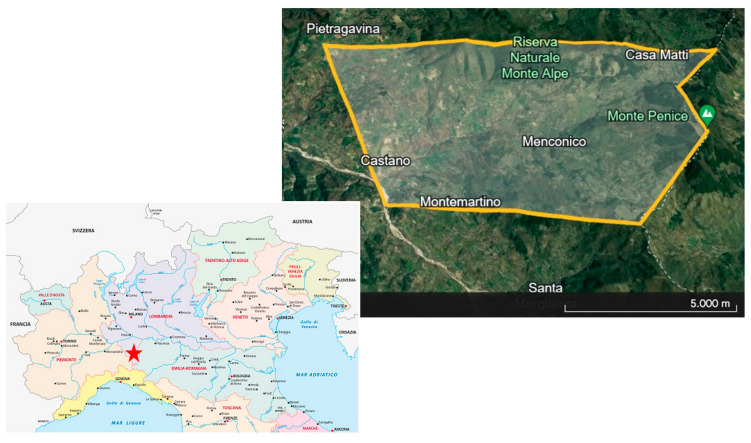
Map of the sampling area. The red star represents the location of the sampling area on the map of northern Italy.

**Figure 2 animals-14-02209-f002:**
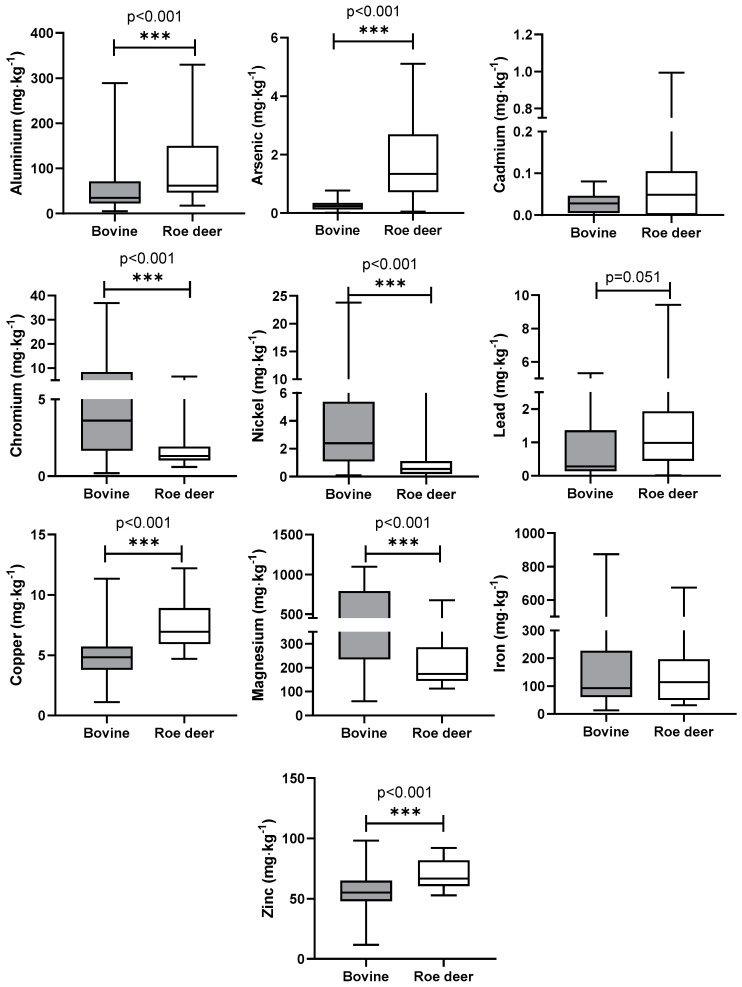
Graphical representation comparing the concentrations of PTEs in the hair of both cattle and roe deer. ***: extremely significant.

**Table 1 animals-14-02209-t001:** ICP-OES device parameters for determination of elements.

Parameters	Assigned Value
Plasma gas flow rate	15 L/min
Argon carrier flow rate	0.5 L/min
Sample flow rate	1.51 L/min
The speed of peristaltic pump	100 rpm
RF Power	1150 W

**Table 2 animals-14-02209-t002:** Wavelengths used in the analysis for each element.

Elements	Wavelength (nm)
Aluminum (Al)	167.070
Arsenic (As)	189.042
Cadmium (Cd)	228.802
Chromium (Cr)	267.716
Nickel (Ni)	341.476
Lead (Pb)	220.353
Copper (Cu)	324.754
Magnesium (Mg)	285.213
Iron (Fe)	259.940
Zinc (Zn)	206.200

**Table 3 animals-14-02209-t003:** Results of ICP-OES method validation for elements.

Elements	Quality Control (QC)	LOD	LOQ	Expected Concentration	Measured Concentration (*n* = 3) (ppm)	Precision (RSD%)	Recovery (%)
Al	QC-1	0.001	0.003	2.500	2.420	0.168	96.8
	QC-2			5.000	5.000	0.144	100
As	QC-1	0.000	0.001	0.050	0.050	5.684	100
	QC-2			0.100	0.100	0.974	100
	QC-3			0.500	0.490	4.248	98
	QC-4			1.000	0.960	1.274	96
Cd	QC-1	0.008	0.014	0.500	0.536	0.752	107.2
	QC-2			1.000	0.963	0.213	96.3
Cr	QC-1	0.003	0.008	0.050	0.052	0.726	104
	QC-2			0.100	0.097	0.816	97
Ni	QC-1	0.000	0.002	0.050	0.052	6.844	104
	QC-2			0.100	0.099	2.474	99
	QC-3			0.500	0.490	0.838	98
	QC- 4			1.000	0.990	0.124	99
Pd	QC-1	0.000	0.004	0.500	0.490	3.332	98
	QC-2			1.000	1.010	0.451	101
Cu	QC-1	0.002	0.006	0.500	0.509	0.32	101.8
	QC-2			1.000	0.990	0.675	99
Mg	QC-1	0.000	0.001	0.500	0.524	1.895	104.8
	QC-2			1.000	0.973	0.539	97.3
Fe	QC-1	0.003	0.004	0.500	0.490	5.772	98
	QC-2			1.000	0.990	0.846	99
Zn	QC-1	0.003	0.009	0.500	0.540	0.756	108
	QC-2			1.000	1.000	0.343	100

QC: quality control; LOD: limit of detection; LOQ: limit of quantitation; RSD: relative standard deviation.

**Table 4 animals-14-02209-t004:** Mean, median, minimum, and maximum concentrations of PTEs quantified in bovine and roe deer hair. The values are expressed in mg·kg^−1^.

				Percentile			
Element	Species	Mean ± SD	Min–Max	25th	Median	75th	*p* Value
Al	Bovine	61.34 ± 71.86	5.26–289.03	22.74	35.00	70.12	<0.001
	Roe deer	98.64 ± 77.03	17.88–329.6480	46.93	61.88	144.04	
As	Bovine	0.26 ± 0.17	0.01–0.77	0.14	0.24	0.34	<0.001
	Roe deer	1.84 ± 1.51	0.05–5.1	0.78	1.34	2.58	
Cd	Bovine	0.03 ± 0.03	0–0.08	0.01	0.03	0.04	0.167
	Roe deer	0.08 ± 0.16	0–0.99	0.00	0.05	0.10	
Cr	Bovine	5.66 ± 6.68	0.19–36.91	1.75	3.62	8.25	<0.001
	Roe deer	1.61 ± 1.04	0.59–6.58	1.04	1.31	1.89	
Ni	Bovine	3.73 ± 4.34	0.09–23.78	1.38	2.39	5.33	<0.001
	Roe deer	0.83 ± 1.13	0–6.78	0.19	0.54	1.05	
Pb	Bovine	0.97 ± 1.39	0–5.32	0.16	0.28	1.30	0.051
	Roe deer	1.39 ± 1.63	0.01–9.42	0.46	0.99	1.92	
Cu	Bovine	4.92 ± 2.2	1.11–11.35	3.84	4.83	5.60	<0.001
	Roe deer	7.49 ± 2.09	4.71–12.21	5.95	6.95	8.84	
Mg	Bovine	490.93 ± 306.01	59.96–1097.08	258.56	384.77	777.63	<0.001
	Roe deer	226.89 ± 120.14	112.24–675.22	147.23	173.84	279.87	
Fe	Bovine	175.13 ± 196.38	13.09–873.56	65.24	93.32	212.01	0.984
	Roe deer	156.11 ± 143.39	0–674.14	50.21	113.77	188.79	
Zn	Bovine	54.43 ± 19.34	11.71–98.12	48.11	55.06	64.90	<0.001
	Roe deer	69.34 ± 11.54	52.78–92.03	60.66	66.79	78.96	

## Data Availability

The data presented in this study are available in the article and Appendix A. Further information is available upon request from the corresponding author. **Aknowledgements:** The authors acknowledge support from the University of Milan through the APC initiative.

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
