# Peer review of "Use of Hair as Matrix for Trace Elements Biomonitoring in Cattle and Roe Deer Sharing Pastures in Northern Italy"

_animals, 2024, doi:10.3390/ani14152209_

Round 1

Reviewer 1 Report

Comments and Suggestions for Authors

The work presented for review deals with an interesting and current topic regarding the possibility of determining trace elements in cattle hair and Roe deer.

The manuscript was prepared correctly, but I have a few comments:

Line 45-54: this fragment is unnecessary. Does not apply to research topics

Line 121: please replace "Arcus costalis" for "costal arch (arcus costalis)

Line 161: please standardize the way of writing units of measurement

Line 195: please standardize the way of writing units of measurement

Line 267: please standardize the way of writing units of measurement

Author Response

Answer to reviewer 1

The work presented for review deals with an interesting and current topic regarding the possibility of determining trace elements in cattle hair and Roe deer.

Dear reviewer 1,

Thank you for the kind reviews, below you can find the answers to your comments.

The manuscript was prepared correctly, but I have a few comments:

Comment 1

Line 45-54: this fragment is unnecessary. Does not apply to research topics

Response 1

Line 47-56: the fragment has been removed.

Comment 2

Line 121: please replace "Arcus costalis" for "costal arch (arcus costalis)

Response 2

Line 125: the term “Arcus costalis” has been replaced with the term “costal arch”

Comment 3

Line 161: please standardize the way of writing units of measurement

Response 3

Line 169: the unit of measurement has been corrected.

Comment 4

Line 195: please standardize the way of writing units of measurement

Response 4

Line 203: the unit of measurement has been corrected.

Comment 5

Line 267: please standardize the way of writing units of measurement

Response 5

Line 274-288: the unit of measurement has been corrected.

Reviewer 2 Report

Comments and Suggestions for Authors

This is a very well-presented manuscript describing a study of value in the field of biomonitoring in domestic and wild animals.

I think it would be helpful to state in the abstract that ICP-OES was the method of analysis.

Could you please clarify how the ages of the roe deer were determined.

Some of the cited references describe washing hair samples in solvents to ensure that the results reflected the elements inside the hair shafts rather than in dust or dirt that was on the hair. Was this done in this study? If so, I think this should be stated. If not, I think the potential for surface contamination should be discussed, given the different texture of hair between the species. 

Author Response

Response to Reviewer 2

This is a very well-presented manuscript describing a study of value in the field of biomonitoring in domestic and wild animals.

Dear reviewer 2,

Thank you for your kind revisions, below you can find answers to your comments.

Comment 1

I think it would be helpful to state in the abstract that ICP-OES was the method of analysis.

Response 1

Thank you for the suggestion, the technique used has been added to the abstract.

Comment 2

Could you please clarify how the ages of the roe deer were determined.

Response 2

The age of the animals was estimated through the evaluation of dental eruption and erosion. A sentence to define the method was also added to lines 122 and 123 of the materials and methods.

Comment 3

Some of the cited references describe washing hair samples in solvents to ensure that the results reflected the elements inside the hair shafts rather than in dust or dirt that was on the hair. Was this done in this study? If so, I think this should be stated. If not, I think the potential for surface contamination should be discussed, given the different texture of hair between the species. 

Response 3

Yes, the hair was washed with acetone, three times with deionised water and finally with acetone to remove surface impurities and any adhesive contamination present on the surface of the hair before starting the procedure. At line 156-158 the washing procedure has been stated.

Reviewer 3 Report

Comments and Suggestions for Authors

Simple Summary and Abstract

Authors stated:” potentially toxic elements (PTEs are essential at low concentrations but toxic at high levels”

I did not agree with this definition, please distinguish between nonessential and essential trace elements; more over authors listed Aluminum, As, Cd, Cr, Ni, Pb, Cu, Mg, Fe, and Zn then they have Mg that is a macro element and arsenic that is a metalloid, Cd, Ni that are nonessential, Cr, Cu, Fe and Zn that are essential trace elements. Please justify the choice of these elements, since ICP technique can allow multielement determinations.

Introduction:

same problem authors wrote “the essential and non-essential elements, also called potentially toxic elements (PTEs)”

Cd is a known toxic, others such as Cu could be toxic at certain levels, please explain.

Materials and methods:

were the hair decontaminated before analysis? In which way? If not the concentrations of elements reflect environmental deposition, if yes, internal burden.

Discussion: “Hair analysis is worldwide considered a suitable method for the assessment of the health status and the mineral metabolism of animals. In this study, ,,,, were quantified in the hair of both cattle and roe deer grazing within  the same geographical area. The aim was to use them as bioindicators for the presence and environmental quantities of PTEs, and subsequently compare the concentrations between the two species to identify any interspecies differences” again you should explain if hair were decontaminated from atmospheric deposition.

Comments on the Quality of English Language

English is fine

Author Response

Response to reviewer 3

Dear Reviewer 3,

Thank you for your kind revisions. Below you can find answers to your comments.

Comment 1

Simple Summary and Abstract

Authors stated:” potentially toxic elements (PTEs are essential at low concentrations but toxic at high levels”

I did not agree with this definition, please distinguish between nonessential and essential trace elements; more over authors listed Aluminum, As, Cd, Cr, Ni, Pb, Cu, Mg, Fe, and Zn then they have Mg that is a macro element and arsenic that is a metalloid, Cd, Ni that are nonessential, Cr, Cu, Fe and Zn that are essential trace elements. Please justify the choice of these elements, since ICP technique can allow multielement determinations.

Introduction:

same problem authors wrote “the essential and non-essential elements, also called potentially toxic elements (PTEs)”

Cd is a known toxic, others such as Cu could be toxic at certain levels, please explain.

Answer to comments 1 about simple summary, abstract and introduction:

Thank you for your valuable feedback. Since the literature, with respect to the elements investigated, proposes many different classifications (essential, non-essential, trace, toxic, heavy metals, metalloids, elements and so on) each time we produce a study of this kind, the choice of definition of these elements is complicated, which is why we particularly appreciate the questions addressed to this topic and the comparison of opinions with respect to the use of one term rather than another.

The locution PTEs was used on purpose, with the belief that potentially toxic elements is the most scientifically correct way of defining elements that are found in both metallic and ionized states. moreover, IUPAC in one of its reports (Pure Appl. Chem, Vol. 74, No. 5, pp. 793-807, 2002) in which strongly advises against using the term heavy in front of metals, also reminds us not to use even the single word toxic,

Toxic metal is an imprecise term (Cd). The fundamental rule of toxicology (Paracelsus, 1493–1541) is that all substances, including carbon and all other elements and their derivatives, are toxic given a high enough dose. The degree of toxicity of metals varies greatly from metal to metal and from organism to organism. Thus, the so called “essential elements” could become toxic at high concentrations (Cu) but are necessary for the regular physiological functions of the organism. Pure metals are rarely, if ever, very toxic (except as very fine powders, which may be harmful to the lungs from whatever substance they may originate). Toxicity, like essentiality, should be defined by reference to a dose–response curve for the species under consideration. This is another term that has been used loosely to refer to both the element and its compounds”

So potentially toxic is quite correct. As for the word metalloid, the IUPAC is not so precise but seems to make no difference between metalloid and semimetal.

For these reasons, many chemists to this day therefore prefer to speak of PTEs and we used this definition.

In the simple summary at line 22-24, in the abstract at line 32-34, and in the introduction at line 65-67, the sentence about PTEs and essential and non-essential elements has been rephrased and clarified.

Our decision to evaluate Aluminum (Al), Arsenic (As), Cadmium (Cd), Chromium (Cr), Nickel (Ni), Lead (Pb), Copper (Cu), Magnesium (Mg), Iron (Fe), and Zinc (Zn) is based on their significant environmental and health relevance. Elements such as Al, As, Cd, Cr, Ni, and Pb are prevalent pollutants with substantial health risks, necessitating their monitoring. Mg, while a macroelement, is included due to its essential role in biochemical processes and environmental exposure assessment. Essential trace elements like Cr, Cu, Fe, and Zn are vital for physiological functions, providing insights into nutrient cycles and environmental impacts on biological systems. The toxicity and regulatory importance of Cd, Ni, Pb, and As further justify their inclusion, given their high toxicity at low concentrations and the need for regulatory monitoring.

While the ICP technique allows for multielement determinations, we focused on these specific elements due to their combined significance in environmental monitoring, health impact, and regulatory guidelines.

Comments 2

Materials and methods:

were the hair decontaminated before analysis? In which way? If not the concentrations of elements reflect environmental deposition, if yes, internal burden.

Response to comment 2

Yes, hair were decontaminated before analysis. A statement which explains the procedure has been added in material and methods at line 156-158:

“before starting the analyses, the hair was washed with acetone, three times with de-ionised water and finally with acetone to remove surface impurities and any adhesive contamination present on the surface of the hair, then.”

Comment 3

Discussion: “Hair analysis is worldwide considered a suitable method for the assessment of the health status and the mineral metabolism of animals. In this study, ,,,, were quantified in the hair of both cattle and roe deer grazing within  the same geographical area. The aim was to use them as bioindicators for the presence and environmental quantities of PTEs, and subsequently compare the concentrations between the two species to identify any interspecies differences” again you should explain if hair were decontaminated from atmospheric deposition.

Answer to comment 3

Thanks for the suggestion, a statement was added in the materials and methods describing the hair washing procedure (lines: 156-158). Thus, the concentration found is independent from the external deposition on the hair.

Reviewer 4 Report

Comments and Suggestions for Authors

Manuscript entitled „Use of Hair as Matrix for Trace Elements Biomonitoring in Cattles and Roe Deer sharing pastures in Northern Italy” presents the results of the determination of trace elements in hairs of roe deer and cows living in Italy. Presented results are complementing the pool of knowledge concerning monitoring of elements content of ruminants.

1 Title of the manuscript is adequate to the its text; “cattles” correct “cattle” or “cows”

2 Keywords: remove trace elements because you have this word in the title of your paper.

3 Abstract: adequate to the content presented in the manuscript.

Introduction:

4 The authors refer to the literature on wild boar and pigs and do not include cervids? Reference should be made to cervids in the introduction.  

5 L 75 „Many wild mammals have been identified as ideal bioindicators…”  Please refer to the literature by inserting a few items, including those concerning of cervids.

Materials and Methods:

6 The investigations were done on sufficient animal material using widely accepted methods. All presented tables and references are necessary and adequate.

Results:

7 L 189 Check the sentence and correct according to the results obtained. It can be seen from the data in Table 4 and Figure 2 that there are no differences in the concentration of Cd in the two species. Analyse the results carefully. The Cd content is not higher in roe deer. Why there is no information on significant differences in zinc ?

Discussion:

8 There are studies available describing minerals including Mg in roe deer hair; please refer to these studies.

10 The sampling period covered autumn. As is known, in early autumn (September - October) of roe deer change their dress from summer to winter. Could the authors refer to the seasonal change of coat ?

Conclusions:

11 The conclusions are appropriate.

References:

12 Journal names should be written in capital letters.

Author Response

Response to reviewer 4

Manuscript entitled „Use of Hair as Matrix for Trace Elements Biomonitoring in Cattles and Roe Deer sharing pastures in Northern Italy” presents the results of the determination of trace elements in hairs of roe deer and cows living in Italy. Presented results are complementing the pool of knowledge concerning monitoring of elements content of ruminants.

Dear reviewer 4,

Thank you for your valuable revisions. Below you can find answers to your comments.

Comment 1

Title of the manuscript is adequate to the its text; “cattles” correct “cattle” or “cows”

Answer to comment 1

The term cattles in title has been corrected in cattle

Comment 2 

Keywords: remove trace elements because you have this word in the title of your paper.

Answer to comment 2

Thank you for the suggestion, the keyword trace elements has been removed.

Comment 3 

Abstract: adequate to the content presented in the manuscript.

Comment 4

Introduction:

4 The authors refer to the literature on wild boar and pigs and do not include cervids? Reference should be made to cervids in the introduction.  

5 L 75 „Many wild mammals have been identified as ideal bioindicators…”  Please refer to the literature by inserting a few items, including those concerning of cervids.

Answer to comment 4

Thank you for the suggestion, some references with studies about cervids has been added. (lines 78-80)

Comment 5

Materials and Methods:

6 The investigations were done on sufficient animal material using widely accepted methods. All presented tables and references are necessary and adequate.

Comment 6

Results:

7 L 189 Check the sentence and correct according to the results obtained. It can be seen from the data in Table 4 and Figure 2 that there are no differences in the concentration of Cd in the two species. Analyse the results carefully. The Cd content is not higher in roe deer. Why there is no information on significant differences in zinc ?

Response to the comment 6

Thank you very much for the suggestions. At line 196 and 197 of the results the list of significance has been corrected.

Comment 7

Discussion:

Response to comment 7

8 There are studies available describing minerals including Mg in roe deer hair; please refer to these studies.

Thank you for your valuable feedback. We have conducted a search for articles on magnesium quantification in roe deer hair and have added the relevant references in lines 314-321.

10 The sampling period covered autumn. As is known, in early autumn (September - October) of roe deer change their dress from summer to winter. Could the authors refer to the seasonal change of coat ?

The sampling period, as reported at line 129, covered the period from June-October, the seasonal change of coat in the area object of study, starts in the second half of October due to the climate characteristics (the autumn is not so cold and the altitude is about 750 meters over sea level). Considering the animals sampled in the last 15 days of October (3) and the possible ongoing coat change, compared to the total number of animals used for this study, we thought it was not a significant data point.

Conclusions:

11 The conclusions are appropriate.

Comment 8

References:

12 Journal names should be written in capital letters.

Answer to comment 8

The references list has been updated.

Round 2

Reviewer 3 Report

Comments and Suggestions for Authors

In my opinion the revised paper is suitable for publication

Comments on the Quality of English Language

none